# Protein Phosphorylation and Redox Status: An as Yet Elusive Dyad in Chronic Lymphocytic Leukemia

**DOI:** 10.3390/cancers14194881

**Published:** 2022-10-06

**Authors:** Mario Angelo Pagano, Federica Frezzato, Andrea Visentin, Livio Trentin, Anna Maria Brunati

**Affiliations:** 1Department of Molecular Medicine, University of Padua, 35121 Padua, Italy; 2Hematology and Clinical Immunology Unit, Department of Medicine, University of Padua, 35128 Padua, Italy; 3Veneto Institute of Molecular Medicine, 35129 Padua, Italy

**Keywords:** chronic lymphocytic leukemia, phosphorylation, redox, antioxidant systems

## Abstract

**Simple Summary:**

Phosphorylation is one of the most crucial modifications of lipids and proteins, as it regulates virtually all cellular functions. Like other human diseases, chronic lymphocytic leukemia (CLL), the most common leukemia in Western developed countries, exhibits deranged phosphorylation, which is induced by stimuli within specific tissues (e.g., lymph nodes), promoting enhanced proliferation and survival. Importantly, a growing body of evidence shows that reactive oxygen species (ROS), altered forms of oxygens generated from metabolism and peculiar enzyme complexes, and generally considered highly harmful due to their reactivity toward critical biomolecules (proteins, lipids, and DNA), act in concert with phosphorylation in supporting the malignant phenotype in CLL. This complex interplay is now providing insights into potential novel Achilles heels of intracellular signals for the development of innovative treatments which might synergize with the drugs currently in use that target the principal players in phosphorylation, namely kinases.

**Abstract:**

Malignant cells in chronic lymphocytic leukemia (CLL) are characterized by oxidative stress that is related to abundant generation of reactive oxygen species (ROS) by increased mitochondrial oxidative phosphorylation (OXPHOS). Lymphoid tissues have been shown to provide a protective microenvironment that antagonizes the effects of ROS, contributing to establishing redox homeostasis that supports the vitality of CLL cells. In the last few decades, a complex antioxidant machinery has been demonstrated to be activated in CLL cells, including the different superoxide dismutase (SOD) isoforms, the thioredoxin (Trx) system, and the enzyme cascade inducing glutathione (GSH) biosynthesis and recycling, to name a few. Their expression is known to be upregulated by the activation of specific transcription factors, which can be regulated by either oxidative stress or phosphorylation. These two latter aspects have mostly been explored separately, and only recently an increasing body of evidence has been providing reasonable inference that ROS and phosphorylation may cooperate in an interplay that contributes to the survival mechanisms of CLL cells. Here, we present an overview of how oxidative stress and phosphorylation-dependent signals are intertwined in CLL, focusing on transcription factors that regulate the balance between ROS production and scavenging.

## 1. Introduction

Chronic lymphocytic leukemia (CLL), the most common adult leukemia in Western developed countries, is characterized by CD5^+^/CD19^+^/CD23^+^ B lymphocytes that proliferate in secondary lymphoid tissues and bone marrow, progressively accumulating in the peripheral blood as mature quiescent cells [1,2]. CLL displays a highly variable clinical course and is still an incurable disease despite the recent remarkable advances in therapy that have improved life expectancy [3]. In this regard, efforts towards novel therapies beyond the standard chemoimmunotherapy have led to the development of small molecules that target factors essential to CLL pathogenesis in malignant B cells. These include Bruton tyrosine kinase (Btk) and phosphoinositide 3-kinase (PI3K), or the anti-apoptotic prototypical member of the B-cell lymphoma 2 (Bcl-2) family, namely Bcl-2 (Bcl-2). These are now used as first-line treatment options [4,5,6,7,8,9]. Importantly, the pathobiology of this disease is also characterized by the pivotal role of the microenvironment within the lymphatic tissues, which is constituted by a variety of non-malignant accessory cells, such as monocyte-derived nurse-like cells (NLCs), T cells and bone marrow stromal cells (BMSCs), that support prolonged survival and proliferation of CLL cells. In confirmation of the importance of this microenvironment, it is a common observation that CLL cells swiftly undergo apoptosis in vitro, unless they are co-cultured with cells that mimic the microenvironment itself, such as leukocytes and stromal cells [10,11,12]. In vivo, these non-malignant cells contribute to the functional characteristics of CLL cells by interaction of their surface-bound ligands, including Cluster of Differentiation (CD) 40 (CD40) ligand, Programmed cell Death protein 1 (PD-1), Vascular Cell Adhesion Molecule 1 (VCAM-1), CD31, or soluble factors such as C-X-C motif chemokine ligand (CXCL) 12 (CXCL12) and CXCL13, Interleukin (IL) 6 (IL-6) and IL-10, B cell Activating Factor (BAFF) and A Proliferation-Inducing Ligand (APRIL), with cognate receptors on the plasma membrane of CLL cells themselves [13,14,15,16]. In this scenario, the most studied receptor that plays a crucial role in the pathogenesis of CLL is the B Cell Receptor (BCR), which sustains the malignant phenotype of CLL cells via antigen-dependent engagement or autonomous autoactivation [17,18]. The signals downstream of BCR activation are then transduced into phosphorylation-driven cascades in which several kinases with abnormally enhanced activity partake. Of these kinases, some are directly associated with the BCR, including Lyn, a Src Family Kinase (SFK) predominantly expressed in B lymphocytes, and Spleen tyrosine kinase (Syk), whereas others, such as Akt, Btk and PI3K are indirectly connected with the BCR through adaptor proteins and modifications of the plasma membrane [19,20]. Significantly, it is to be underscored that the action of the overactive kinases is not properly counterbalanced owing to the repression of the activity and/or expression of a considerable number of phosphatases, some of which will be listed below. There is now compelling evidence that an additional factor that elicits the activation of diverse survival pathways in CLL, as in other blood malignancies, is a perturbed redox balance due to excess reactive oxygen species (ROS), which may ultimately facilitate disease progression and confer drug resistance [21,22]. ROS, once upon a time dismissed as mere by-products of multiple metabolic pathways with the harmful potential to cause damage to cellular macromolecules because of their strong oxidizing potential, are now also being acknowledged as crucial factors in several cellular functions under physiological and pathological conditions such as cell signaling and metabolism [23,24]. In CLL cells, mitochondria, in addition to functioning as “powerhouses” that fulfill energy demand, are also the main source of ROS. These act as signaling molecules by furthering mitochondrial biogenesis to meet metabolic needs and contribute to their own detoxification by inducing the transcription of target genes, mainly mediated by the transcription factor nuclear factor erythroid 2-related factor 2 (Nrf2) [25,26,27,28].

In this review, we summarize recent evidence that highlights phosphorylation and redox homeostasis as crucial processes affecting one another and contributing to the pathogenesis of CLL, with an eye to the redox status as a potential new target for therapeutic intervention.

## 2. A Quick Glance at Phosphorylation-Dependent Signaling in CLL

To gain a general understanding of phosphorylation-mediated signals in CLL cells, it is worthwhile to take an overview of the mechanisms regulating the signaling pathways in normal B cells. B cells sense the microenvironmental conditions through several surface receptors, which in turn transduce such extracellular cues in signaling pathways that ultimately govern the cell response and the cell fate [29]. As previously mentioned, the most investigated receptor of B cells is the BCR, which is involved not only in the immune response but also in the activation of signaling pathways regulating survival, maturation, and migration of the B cell itself. The BCR is composed of an immunoglobulin that recognizes antigens, and the co-receptors CD79a and CD79b that represent its signaling components [17]. Upon engagement of the BCR, the signaling cascade is initiated by the activation of the tyrosine kinase Lyn, which phosphorylates specific tyrosines within the immunoreceptor tyrosine-based activation motifs (ITAMs) of a variety of co-receptors, CD79a and CD79b themselves, and CD19, to name a few [30]. Phosphorylated ITAMs in turn provide docking sites for the Src Homology 2 (SH2) domain-based recruitment, and subsequent activation, of the effector kinases Syk and PI3K, in particular, thereby propagating phosphorylation-mediated signals [30]. In particular, PI3K catalyzes the conversion of phosphatidylinositol (4,5)-bisphosphate (hereafter PIP_2_) into PI(3,4,5)-trisphosphate (hereafter PIP_3_), which provides a platform for the recruitment of enzymes harboring pleckstrin-homology (PH) domains, including Btk, phospholipase Cγ (PLCγ), 3-phosphoinositide-dependent protein kinase-1 (PDK1), and Akt [31,32,33,34] (Figure 1). PLCγ in turn hydrolyzes PIP_2_ into diacylglycerol, an activator of Protein Kinase C (PKC), and inositol 1,4,5-trisphosphate (IP_3_), which mobilizes Ca^2+^ from the intracellular stores [35]. In addition to phosphorylating ITAMs, Lyn uniquely phosphorylates Immunoreceptor Tyrosine Inhibitory Motifs (ITIMs) of inhibitory cell surface co-receptors such as CD22, CD72, and FcγRIIB, which provide docking sites for the SH2 domains of phosphatases, such as Src homology region 2 domain-containing phosphatase-1 (SHP-1) and Src homology 2 (SH2) domain-containing inositol 5′phosphatases (SHIPs), which abolish signaling, eventually contributing to a fine balance between positive and negative signaling pathways [17,36]. There is general agreement that some of the key molecules described above such as Lyn, Syk, PKC, and PI3K are constitutively active in CLL cells, resulting in tonic, ligand-independent BCR signaling [19]. Furthermore, there is substantial evidence that the main actor in this aberrant signaling network is Lyn, which, in addition to being situated beneath the plasma membrane in the close proximity of the BCR, is also found as part of a multiprotein complex aberrantly located in the cytoplasm [37], where it contributes to the phosphorylation of a myriad of substrates implicated in B cell proliferation [38], anti-apoptotic mechanisms [39], and cytoskeletal rearrangement [40,41]. Notably, it has been observed that the elevated level of phosphorylation in CLL cells can be accounted for by the impaired expression or activity of a significant number of protein or lipid phosphatases, including protein tyrosine phosphatase receptor type O (PTPROt) [42], PH Domain and Leucine Rich Repeat Protein Phosphatase 1 (PHLPP1) [43], Src homology (SH) 2 (SH2) domain containing inositol polyphosphate 5-phosphatase 1 (SHIP1 [44,45], Phosphatase and tensin homolog (PTEN) [46,47], and Protein Phosphatase 2A (PP2A) [38,48]. By contrast, Src homology region 2 domain-containing phosphatase-1 (SHP-1) is expressed in CLL at levels comparable to those in normal B cells, occurring in an active form bound to the receptor CD5, and in an inhibited conformation in the cytosol [49,50].

## 3. ROS and Antioxidant Response

Reactive oxygen species (ROS) are molecules generated by the partial reduction of oxygen that include radical species such as superoxide anion (O^−^∙) and hydroxyl radical (OH∙), as well as non-radical species such as hydrogen peroxide (H_2_O_2_) and singlet oxygen [28,51,52,53]. Until recently dismissed as mere by-products generated by multiple metabolic pathways with detrimental effects on cellular macromolecules, ROS are now being also acknowledged as crucial factors in several cellular functions under both physiological and pathological conditions such as cell signaling and metabolism [54]. Some of the most important sources of ROS in biological systems include NADPH oxidases (NOXs, multiprotein complexes located within the biological membranes at the cell surface and cellular organelles), the mitochondrial electron transfer chain, and metabolic enzymes in the endoplasmic reticulum as well as peroxisomes [55].

To cope with the hazardous effects of ROS, cells establish “redox homeostasis”, which is a condition that prevents an imbalance between ROS formation and detoxification [56]. Importantly, this latter process is brought about by an antioxidant machinery composed of factors whose expression can be even induced by ROS themselves, ultimately promoting a negative feed-back loop that “buffers” excess ROS [57,58,59].

Major antioxidant systems include several enzymes that can act either by turning ROS into less harmful forms for further detoxification, or by regenerating the oxidized forms of target molecules. In this regard, Superoxide Dismutases (SODs) serve as an early defense response that convert the basic form of ROS, namely O^−^∙, to the less reactive H_2_O_2_. There exist three different isoforms of SOD, SOD1 (Cu/Zn SOD) being distributed over a wide range of subcellular compartments, SOD2 (Mn-SOD) being localized in the mitochondria, and extra-cellular (EC) SOD [60]. In more detail, SOD2 detoxifies O^−^ originating from the reaction of electrons leaking from the respiratory chain with oxygen, whereas EC-SOD cooperates with NOXs at the plasma membrane for generating H_2_O_2_, which reenters the cell to act as signaling mediator [61]. H_2_O_2_ can be in turn reduced to water and oxygen by catalase [62,63] or to water by glutathione (GSH) peroxidase (GSH-Px, hereafter GPx) [64,65]. Neutralizing H_2_O_2_ is crucial for the cell’s integrity because H_2_O_2_ in the presence of reduced metals can give rise to OH∙, a highly reactive hydroxyl radical that peroxidizes organic molecules in the immediate vicinity [66]. One of the most dangerous effects is lipid peroxidation, which can bring about the breakdown of polyunsaturated fatty acid and the formation of highly reactive aldehydes, which in turn react with specific residues in proteins, cysteines and lysines among others, eventually compromising the function of the proteins themselves [67].

As to cysteines within proteins, whose thiol groups are prone to oxidation, these residues can be reduced back to their original form by thioredoxins (Trx1 and Trx2), at the expense of their own thiols, which are again reduced by action of thioredoxin reductases (TR1, 2 and 3) [68] and glutaredoxins [69]. For the whole antioxidant machinery to be efficient and cellular thiols to be preserved, substantial amounts of GSH, the cofactor of several antioxidant enzymes, must be either recycled by reduction, mediated for instance by glutathione reductase (GR) [70], or synthesized ex novo [71]. GR requires NADPH as a cofactor, which is principally provided by glucose 6-phosphate dehydrogenase (G6PDH) and 6-phosphogluconate dehydrogenase (6PGDH) in the pentose phosphate pathway (PPP) [72], whereas γ-glutamyl cysteine ligase (GCL) is the rate-limiting enzyme in GSH biosynthesis followed the action of GSH synthetase [71].

Other enzymes that are upregulated by oxidative stress are Glutathione S-transferase (GST), a phase II enzyme that detoxifies electrophilic compounds and contributes to repairing oxidation-related cell damage [73], NAD(P)H dehydrogenase [quinone] 1 (NQO1), whose activity as a quinone reductase prevents the generation of radical species [74], and heme oxygenase 1 (HO-1), whose protective action is due to its ability to catalyze heme degradation, acting as a chaperone in the regulation of vital signaling pathways [75] (Figure 2). Interestingly, whereas the unrestrained increase in ROS levels and the inability to restore the redox homeostasis has been recognized to disrupt the structure and function of virtually all target biomolecules and underlie tumorigenesis [76], malignant cells themselves have been shown to massively develop antioxidant systems to counter the potential damages from ROS generated by their elevated metabolic demands [77,78].

CLL cells, in contrast with other blood malignancies that produce ROS through enhanced activity of NOXs, with a metabolic shift to aerobic glycolysis (“Warburg effect”) [79], have been shown to express low levels of the catalytic subunit (gp91phox) of NOX2, the most representative NOX2 in myeloid and lymphoid cells [25], displaying increased oxidative phosphorylation (OXPHOS) [21,80]. The substantial electron leakage that results from elevated OXPHOS induces oxidative stress and subsequent mitochondrial damage, which triggers the expression of factors, such as of HO-1 [81], that promote de novo mitochondrial biogenesis [22], and enzymes directly involved in the GSH synthesis, such as GCL and G6PDH, resulting in a higher amount of thiols relative to normal B cells [26,82]. Interestingly, the data regarding catalase and SOD2 are still controversial, although most authors agree on a reduced expression of the former and increased activity of the latter [83,84,85,86].

Importantly, the “negative feed-back loop” that detoxifies ROS and that is stimulated by ROS themselves illustrated above occurs via the activation of transcription factors, including nuclear factor erythroid 2-related factor 2 (Nrf2), the nuclear factor kappa-light-chain-enhancer of activated B cells (NF-κB), and Forkhead box O transcription factors (FOXO), which are regulated, directly or indirectly, by oxidation, by phosphorylation or both.

## 4. Nrf2, the Master Regulator of Antioxidant Responses: A Complex Regulation for Fine-Tuned Redox Homeostasis

Nrf2 is a pivotal transcription factor composed of seven conserved Nrf2-ECH homology (Neh) domains, which, in response to oxidative stress and toxic insults, induces the expression of a wide variety of proteins actively involved in the antioxidant response [87,88], as exemplified in Figure 3A. In CLL, it has been demonstrated that Nrf2 is highly expressed and is key to survival, as corroborated by studies highlighting the cytotoxic effects of electrophilic and antioxidant compounds targeting Nrf2 signaling [26]. Nrf2 signaling orchestrates a robust and varied response to the abundant mitochondrial production of ROS in CLL cells, especially promoting the overexpression of GCL modulatory and catalytic subunits contributes to maintaining a high content of GSH [89] as well as the upregulation of HO-1, a positive regulator of transcription factor A, mitochondrial (TFAM), which stimulates mitochondrial biogenesis as a mechanism compensating for the ROS-induced damage and decreased energy production of mitochondria [90].

Nrf2 expression and activity are strictly regulated by several mechanisms, including post-translational modifications, especially phosphorylation, as well as protein-protein interaction, miRNAs and the cell’s redox balance [91]. Under basal conditions, Nrf2 remains sequestered and inactive in the cytoplasm, being bound through the Neh2 domain to Kelch-like ECH-associated protein 1 (Keap1), an adaptor protein that ensures Nrf2 proteasomal degradation via interaction with the Cullin3 (Cul3)-containing E3-ligase complex [92]. To act as a transcription factor, Nrf2 must be first released from the complex and accumulate in the nucleus, where it forms heterodimers with other transcription factors, especially small Musculoaponeurotic fibrosarcoma proteins (sMafs), thereafter binding to the so-called Antioxidant Response Elements (AREs) and triggering the transcription of its target genes [93]. Diverse mechanisms contribute to the disruption of the Nrf2-Keap1-Cul3 complex (Figure 3B). Keap1, for instance, senses alterations of the redox balance through its reactive cysteines, which are swiftly oxidized under oxidative stress, with conformational changes that eventually allow Nrf2 to be released [93]. An additional mechanism involves the interaction of specific molecules with Keap1, ultimately liberating Nrf2 by competition and stabilizing it. One such factor is the cyclin-dependent kinase inhibitor p21, a p53-target gene, which has been shown to partake in a positive cooperation between the Nrf2- and p53-dependent pathways for mitigating oxidative stress [94]. Another protein that acts in a manner similar to p21, and that is otherwise involved in autophagy, is p62, which contributes to countering oxidative stress in a signaling loop that consists of p62 itself, Nrf2 with NF-kB [95], as is described in more detail below in relation to NF-kB.

Nrf2 can also be regulated through phosphorylation of its individual domains mediated by several protein kinases, including some isoforms of the PKC family, Glycogen Synthase Kinase 3 (GSK3), AMP-activated protein Kinase (AMPK), Casein Kinase 2 (CK2) and Cyclin Dependent Kinase 5 (CDK5) [96]. Herein we focus on PKC and GSK3 because (a) they have been shown to play a key role in regulating the level of expression and the entry into the nucleus of Nrf2 itself, and, in addition, in an opposite manner, as described below, and (b) their level of activity is dysregulated in CLL, which makes it reasonable to infer that the activation status of these protein kinases themselves can affect Nrf2 signaling.

The different PKC isoforms share a highly conserved catalytic domain, with a variable regulatory domain by which three distinct subfamilies can be subdivided according to their mechanism of activation, namely the so-called classical Ca^2+^- and diacylglycerol-dependent isoforms (α, βI/II, and γ), novel diacylglycerol-dependent and Ca^2+^-independent isoforms (δ, ε, η, and θ), and atypical isoforms (ζ, and ι/λ) [97,98]. Interestingly, PKC activity has been also found to be elicited by oxidative stress, which can induce oxidation of reactive cysteines within the Zn^2+^ finger domain, thereby disrupting the autoinhibitory interactions [97]. In this regard, the PKC family could be considered a sensor of elevation of ROS and as a trigger for a feed-back signaling loop aimed at attenuating the effects of oxidative stress through Nrf2 upregulation. Indeed, this view is supported by the evidence that, although it is still unclear which PCK isoform is involved, the Nrf2 Neh2 domain (residues 1–86) undergoes PKC-dependent phosphorylation at Ser40, thereby inducing dissociation of Nrf2 from Keap1 with subsequent translocation of Nrf2 itself into the nucleus and activation of the ARE-mediated antioxidant response [99]. Conversely, GSK3, a constitutively active key molecular switch in metabolism and signal transduction under cellular resting conditions, negatively regulates Nrf2 directly by phosphorylating it or indirectly by promoting Nrf2 phosphorylation via activation of the Src Family Kinase Fyn. In the former case, GSK3 phosphorylates Ser335 and Ser338 in the Neh6 domain, promoting the interaction with a redox-independent ubiquitin E3 ligase adaptor, β-TrCP, resulting in KEAP1-independent proteasomal degradation of Nrf2 [100]. Regarding the inhibitory mechanism mediated by the GSK3/Fyn axis, GSK3 phosphorylates Fyn, which can thereafter translocate into the nucleus, in turn phosphorylating Nrf2 at Tyr568, which causes Nrf2 nuclear exclusion [101]. Importantly, GSK3 is functionally located downstream of the PI3K/Akt signaling axis, which, when activated under either physiological and pathological stimulation, brings about phosphorylation of N-terminal serines (Ser21 and Ser9 of the GSK3α and β isoform, respectively), resulting in the inhibition of GSK3 itself [102]. It is to be noted that oxidative stress contributes to the loss of function of critical players that deactivate the PI3K/Akt axis, such as the phosphatases PTEN [103] and PP2A [104], strongly contributing to the overactivation of Akt and subsequent GSK3 inactivation.

All the above, although no data are thus far available on the phosphorylation status of Nrf2 in CLL, aid in elaborating a model that involves PKC and GSK3 in light of the altered signaling network in this disease. Indeed, the activity of these two protein kinases is strikingly unbalanced in favor of PKC, which is constitutively active and situated downstream of the tonic signals originating from the “signalosome” underneath the BCR, thereby imparting survival and proliferative cues in malignant cells [105]. It cannot be ruled out that PKC sustains CLL cell survival also by partaking in Nrf2-mediated signaling by phosphorylating Ser40, as described above. On the other hand, GSK3 is kept inactive by a phosphorylation-dependent mechanism mediated by the PI3K/Akt axis, which is constitutively active in CLL, as in several other blood malignancies [106]. This inhibition is further supported by PP2A activity impairment, which would otherwise dephosphorylate, and thus activate, GSK3. In this scenario, it can be hypothesized that GSK3 inhibition, as PKC activation, are parallel mechanisms that reinforce Nrf2 translocation to the nucleus and transcriptional activity upregulated by oxidative stress.

However, further research is warranted to determine whether such mechanisms, as observed in other types of cancer, characterize CLL and may represent novel targets for countering CLL cell antioxidant defenses.

## 5. NF-kB, Multifunctional Complexes for Pleiotropic Actions upon Oxidative Stress

Nuclear Factor kappa-light-chain-enhancer of activated B cells (NF-kB) is a family of transcription factors that induce the expression of myriad gene targets implicated in immune response, inflammation, cell growth, cell survival, apoptosis and tumorigenesis [107]. In CLL cells, the expression and activity of the NF-kB members, which are higher than in non-malignant B cells, are closely related to the factors released by the microenvironmental cells, thereby enhancing survival and proliferation [108].

These factors occur as homo- and heterodimers consisting of one of the Rel family members, namely RelA (p65), RelB and c-Rel, with the protein partners NF-κB1 (p50) and NF-κB2 (p52) [109]. Although the various NF-κB dimers regulate the expression of similar sets of genes, each NF-κB subunit combination promotes specific expression signatures that reflect variable affinity to specific cognate DNA sequences and transactivation activity, this latter being also finely modulated by post-translational modifications including phosphorylation and oxidation state [110,111]. The most common dimeric complexes are p65/p50 and RelB/p52, which are involved in the distinct signaling cascades referred to as canonical and non-canonical pathways, respectively. In brief, NF-kB dimers are located in the cytoplasm under cellular resting conditions owing to two different mechanisms, p65/50 being associated with Inhibitor of NF-kB (IkB), especially IκBα, and RelB being complexed with p100, the precursor of p52. As to the canonical pathway, the activation of the p65/p50 dimer and the consequent translocation into the nucleus are induced by degradation of the different forms of IkB, which is in turn promoted by phosphorylation-dependent ubiquitination. Responsible for IkB phosphorylation is IkB kinase (IKK), an enzyme complex comprised of two catalytic subunits, IKKα and IKKβ in complex with the modulatory subunit NF-kappa-B essential modulator (NEMO or IKKγ) [112], whose activity is elicited by phosphorylation events that in turn are accounted for by ligation of cytokine receptors, Toll-like receptors (TLRs), the BCR and the T cell receptor, to name a few [113]. Other receptors such as the lymphotoxin-beta receptor (LTβR), CD40 and the B cell-activating factor receptor (BAFF-R) trigger the noncanonical pathway, which entails the activation of NF-κB-inducing kinase (NIK) that phosphorylates and activates a IKKα dimer, which in turn phosphorylates p100, eventually inducing the cleavage of p100 itself into p52 [112].

In addition to phosphorylation as a major mechanism of regulation, mounting evidence has recently highlighted a role for ROS in either activating or restraining NF-κB signaling, in a cross-talk with the phosphorylation-associated events related to the canonical and non-canonical pathways [111,114]. ROS can directly oxidize cysteine residues of NF-kB subunits, as in the case of p50 at Cys62, hindering DNA binding [115,116], or indirectly influence NF-kB transcriptional activity by stimulating phosphorylation, as is the case of RelA being phosphorylated at Ser276 by Protein Kinase A (PKA) [117]. However, the prototypical target in NF-kB signaling, whose function may be profoundly affected by ROS in opposite ways, is the IKK complex. Indeed, whereas severe oxidative stress abrogates IKKβ activity via oxidation at Cys179 [118], the regulatory subunit NEMO can undergo oxidation of Cys54 and Cys347, with subsequent formation of a disulfide bond and dimerization that ultimately triggers IKK activity [119]. Interestingly, ROS can also induce phosphorylation-mediated activation of the NF-kB pathway without IκBα proteasomal degradation [120,121]. In this regard, an increase in ROS results in activation of SFKs, such as Src, inducing phosphorylation of IκBα at Tyr42, which is further reinforced by concurrent inactivation of tyrosine phosphatases via oxidation of redox-sensitive thiols in their active sites [122]. As a result, Tyr42 phosphorylation triggers the interaction of IκBα with the PI3K regulatory subunit p85, eventually facilitating the release of the active NF-kB dimer [123]. It is worth mentioning that a similar mechanism of NF-kB activation might be ascribed to the phospho-Tyr42-mediated interaction of IκBα with Src itself [124]. Conversely, high levels of ROS can inactivate the NF-κB cascade by oxidation and glutathionylation of Tyr189 of IκBα, which cannot be further phosphorylated and undergo degradation [125] or by inhibition of the proteasome [126].

Besides being influenced by the redox status of the cell, NF-kB has a role in attenuating the effects of ROS elevation by enhancing expression of target genes, including SOD2 [127,128], Trx1 and Trx2 [128,129], and HO-1 [130].

In CLL cells, a substantial body of evidence demonstrates the upregulation of NF-kB target genes, especially when malignant cells are exposed to stimuli of the microenvironment of lymphoid tissues that promote survival, proliferation, chemotaxis and drug resistance [131]. Both the NF-kB canonical and noncanonical pathways are constitutively active in CLL cells, unfolding upon engagement of extracellular cues to cognate receptors, such as Toll-like receptors (TLRs) [132], CD40 [13], BAFF-R, B-cell maturation antigen (BCMA), Transmembrane Activator and Calcium-modulating cyclophilin ligand (CAML) Interactor (TACI) [16], not to mention the role of protein kinases downstream of the BCR, especially PKC [131]. Other mechanisms of NF-κB activation in CLL that have recently come into the limelight include an unexpected role for BTK, which has been found to directly interact with and to phosphorylate IκBα, which does not undergo degradation [133], and the engagement of the microenvironmental factor Wnt5a with Receptor tyrosine kinase-like Orphan Receptor 1 (ROR1), an oncoembryonic orphan receptor, resulting in an autocrine signaling loop triggered by Signal Transducer And Activator Of Transcription 3 (STAT3) via NF-kB-mediated expression of IL-6 [134]. Thus far, most of the studies illustrating the NF-kB gene signatures in this disease have underscored the overwhelming expression of pro-inflammatory cytokines and anti-apoptotic molecules, rather than antioxidant proteins, as in other cancer types [135]. Nevertheless, a recent study highlighted that NF-κB signaling is part of a crosstalk with the master regulator of the antioxidant response Nrf2 [136]. In brief, BAFF-engaged receptors activate NF-kB signaling with the expression of several genes, one of which is p62, an adapter molecule with a role in autophagy [137]. This protein relieves the Keap-1-mediated inhibition of Nrf2 by competition, promoting the expression of Nrf2-regulated genes coding for p62 itself, NQO1 and HO-1 in particular [136]. This signaling network accounts for enhanced proliferation, chemotaxis, survival and drug resistance of CLL cells with high levels of ROR1, as well as for shorter median treatment-free survival relative to patients with lower levels of expression of ROR1 [138]. Intriguingly, recent findings suggest that the ROS-scavenging activity of the p62-NRF2 axis can be circumvented by exploiting the ability of NQO1 to activate bioreductive prodrugs, in particular one named 29h, which then induces apoptosis, showing that novel targets related to the redox status of cancer cells might open promising avenues for cancer treatment [136].

## 6. The FOXO Family, the Paradox in the Struggle against Cellular Stress and Oxidation

As illustrated in the two examples above, CLL cells hijack signaling pathways that counter the potential harmful effects of ROS in order to ensure the integrity of the molecular mechanisms underlying survival, metabolism and proliferation. Yet, paradoxical as it may seem, it has also been demonstrated that transcription factors promoting genes capable of neutralizing ROS are downregulated in this disease as well as in other forms of cancer. An epitome of this condition is a family of proteins called Forkhead box O (FOXO), the four isoforms of which (FOXO1, FOXO3, FOXO4, and FOXO6) are known to play a key role in cell differentiation, cell cycle arrest, senescence and apoptosis as well as the cellular stress response and antioxidant defense [139,140,141]. Importantly, studies conducted in vitro and on knock-out animal models has led to the assumption that FOXOs can be regarded as tumor suppressors [142]. Interestingly, studies conducted on freshly isolated CLL cells have shown upregulation of FOXO1 and FOXO4, FOXO3 expression being mostly comparable to that in normal B cells [143].

In response to oxidative stress, FOXO proteins promote the expression of SOD2, catalase and GPx-1 [144], the action of which has been described above, as well as different isoforms of peroxiredoxins (Prxs), especially Prx3, and Prx5, which catalyze the reduction of H_2_O_2_ and alkyl hydroperoxides to water and the corresponding alcohol, and Trx2 and TrxR2, which reduce and reactivate the oxidized forms of Prxs and Trxs, respectively [145]. Location and activity of FOXOs are tightly regulated by post-translational modifications, especially cycles of reversible phosphorylation and acetylation, which are brought about by changes in the intra- and extracellular environment, such as membrane receptor engagement (e.g., insulin) or ROS elevation, respectively [146]. Akt is the prototypical protein kinase downstream of transmembrane receptor activation that phosphorylates, thereby disrupting the function of FOXOs [147]. Mechanistically, specific Akt-phosphorylated FOXO residues (T32 and S253 of FOXO3A) turn into binding sites for 14-3-3 proteins, resulting in FOXO shuttling from the nucleus to the cytoplasm with abrogation of transcriptional activity [148]. Other protein kinases involved in the inhibition of FOXOs, eliciting the same or other mechanisms, are Casein Kinase 1 (CK1), Dual Specificity Tyrosine Phosphorylation Regulated Kinase 1A (DYRK1A), IKK and Extracellular-regulated kinase (Erk) [149]. On the other hand, FOXO phosphorylation triggered by mild oxidative stress and mediated by c-Jun N-terminal kinase (JNK) at T447 and T451 and Mammalian Ste20-like kinase (MST1) at S207 increases FoxO activity and nuclear localization or causes FOXO relocation to the nucleus by interfering with the FOXO/14-3-3 interaction, augmenting antioxidant gene expression [147].

As to acetylation, it is promoted by the cAMP response element-binding protein (CREB)-binding protein (CBP) /p300 complex, which acts as a histone acetylase and a coactivator of numerous transcription factors [150]. It has been shown that FOXO acetylation at specific lysine residues weakens binding to DNA, thereby attenuating FOXO activity, a condition that is strongly fostered by oxidative stress. This latter is particularly evident in the case of FOXO4 which, when oxidized at specific cysteines, binds to CBP/p300 via formation of a disulfide bond [151]. Interestingly, recent evidence suggests that acetylation is a precondition for Akt-mediated phosphorylation, in a synergism that further curbs FOXO activity [152]. These inhibitory mechanisms are countered by the NAD-dependent sirtuin (SIRT) 1 (SIRT1) and SIRT2, which decisively contribute to acetylation and dephosphorylation with functional restoration of FOXOs [150,153].

As mentioned above, FOXO proteins count among tumor suppressors and have been shown to be functionally downregulated in many human cancers [141]. In CLL, the microenvironment of secondary lymphoid organs strongly contributes to FOXO3a inhibition, which enhances CLL cell survival. In fact, nurse-like cells secrete factors including chemokines, especially CCL12, that engage C-X-C chemokine receptor type 4, the most prominent chemokine receptor in CLL cells, activating a pathway that involves Akt, which in turn phosphorylates downstream substrates including FOXO3a [140]. As expected, FOXO3a binds to 14-3-3, undergoing nuclear exclusion and functional impairment [48]. Intriguingly, FOXO3a inhibition in CLL cells does not seem to affect the expression of SOD2, which conversely exhibits increased activity as compared to that measured in normal B cells. It can be surmised that other transcription factors compensate for FOXO3a inhibition with regard to SODs, which instead is not the case of catalase, whose levels remain low, supporting the hypothesis that its expression is tightly linked to FOXO3a itself [86]. Notably, it has been hypothesized that high levels of H_2_O_2_ resulting from low catalase expression might be instrumental in the activation of survival pathways, e.g., the PI3K/AKT/mTOR axis, again emphasizing the importance of ROS in a redox balance that maintains a malignant phenotype [86].

## 7. ROS as Candidates for Therapeutic Intervention

It should be reiterated that ROS are essential for cell physiology, unless they increase in an uncontrolled manner or are not substantially scavenged by cellular antioxidant systems. Indeed, the consequent oxidative stress may contribute to the alterations of the cell biology, jeopardizing cell life or driving cell transformation. Thus, the role of oxidative stress in cancer poses the question whether ROS-targeted therapies can be developed. In the past decades, the claim that dietary supplements, e.g., vitamins C and E, β-carotene and selenium, to name a few, might serve as a preventive or curative measure applicable to cancer or other diseases, has been challenged by numerous studies, raising the possibility these supplements could even have detrimental effects [154,155,156,157]. Some authors surmise that the failure of this approach may reflect the inability of these nutrients to reach an effective therapeutic concentration at the site of production and to inactivate ROS. Others hypothesize their interference with crucial important ROS-mediated cellular processes [158,159,160], thereby triggering signals that even support oncogenesis or prevent apoptosis of transformed cells [161]. On the other hand, it should be noted that cancer cells keep the elevation of ROS under control by hijacking the signaling pathways that regulate the antioxidant response, thereby creating a new redox balance that ultimately supports growth, metabolism, survival, and even chemoresistance. Thus, it is not paradoxical that considerable attention is now focused on new treatments utilizing agents that can induce accumulation of ROS and overwhelm the antioxidant capacity of the tumor cell, ultimately leading to cell death. Worth mentioning are agents such as brusatol [162], halofuginone [163], and K67 [164], which directly counters “Nrf2 addiction” [165] by promoting Nrf2 degradation by multiple mechanisms, thereby sensitizing cancer cells to chemotherapeutics, or mitomycin C, which exploits the NRF2-dependent gene products for its bioactivation and is already approved for clinical use [165,166,167].

As for CLL cells, a number of studies supports the view that there exist several compounds capable of inducing cell death by overwhelming the antioxidant defenses through ROS overproduction, especially by targeting ROS-scavenging enzymes. This is the case of ellagic acid, a polyphenolic compound, and acacetin, a natural flavone, which selectively target mitochondria and induce ROS formation [168,169]; highly similar effects are produced by 2-methoxyestradiol (2-ME) and β-phenylethyl isothiocyanate (PEITC), albeit through different mechanisms, the former by inhibiting superoxide dismutase [83], the latter by inducing swift depletion of GSH [170].

Recently, a promising approach has involved oridonin-derivatives, which act as pro-oxidants upon activation by NQO1, which is abundantly expressed in CLL cells overexpressing the orphan receptor ROR1. One of these compounds, 29 h, has been also shown to restore and potentiate the mitochondrial oxidative stress brought about by venetoclax [136], a Bcl-2-directed drug currently used alone or in combination with anti-CD20 monoclonal antibody for relapsed/refractory in the treatment of CLL [171], to which ROR1-rich CLL cells exhibit low sensitivity [139]. It should be remarked that ibrutinib (the first-in class BTK inhibitor) and venetoclax itself, which are currently used in the treatment of CLL, have been shown to trigger ROS production by mechanisms that have not yet been fully elucidated. Ibrutinib-treated CLL cells display higher ROS compared to untreated cells, which results in substantial inhibition of several protein phosphatases, including the B cell receptor signaling regulators SHP-1 and SHIP1 [172]. As to venetoclax, it cannot be ruled out that, as already demonstrated for other leukemias, it can promote dissociation of the Nrf2/Keap-1 complex and target Nrf2 to ubiquitination and proteasomal degradation [173], which would contribute to thwarting antioxidant defenses of CLL cells.

## 8. Conclusions

Phosphorylation-dependent signals have so far been considered key determinants of the malignant phenotype in CLL, with microenvironmental cues playing a crucial role by engaging a variety of CLL cell receptors in the lymphoid tissues, including the BCR. Still, although there is no conclusive evidence, novel insights into the mechanisms supporting B cell survival and proliferation suggest that ROS might be central to these processes. ROS in CLL are predominantly generated in the mitochondria (in contrast to other tumors where NOXs exert this function), and their levels are properly “buffered” by the action of antioxidant factors, whose expression is under the control of transcription factors that are widely known to be regulated by oxidation and phosphorylation. Yet, the interplay of these latter conditions in CLL needs further exploration, especially to establish the role of oxidative stress on the activation status of kinases and phosphatases more precisely. Such investigations might open up new prospects not only for the identification of yet unidentified signaling crosstalks that tie them, but also provide alternative avenues to treatment.

## Figures and Tables

**Figure 1 cancers-14-04881-f001:**
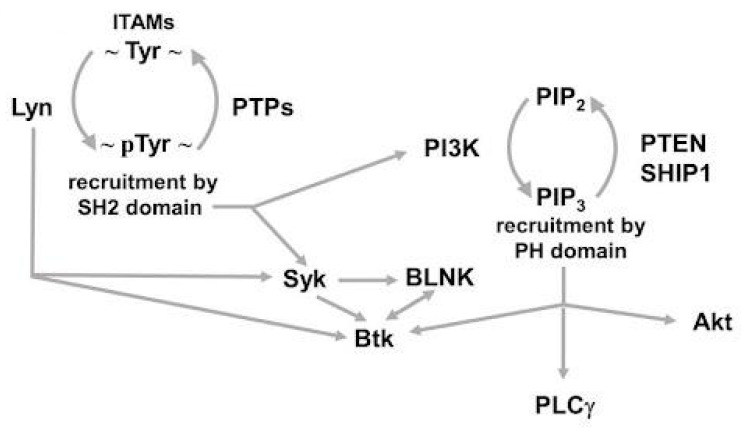
Schematic representation of the main events and mechanism of activation in the “signalosome” downstream of BCR. ITAM, immunoreceptor tyrosine-based activation motifs; pTyr, phosphotyrosine; PTP, protein tyrosine phosphatase; PI3K, phosphoinositide-3-kinase; PIP2, phosphatidylinositol (4,5)-bisphosphate; PIP3, PI(3,4,5)-trisphosphate (PI(3,4,5)P3); PTEN, Phosphatase and tensin homolog; SHIP1, Src homology 2 (SH2) domain containing inositol polyphosphate 5-phosphatase 1; PLCγ, phospholipase Cγ; BLNK, B cell linker.

**Figure 2 cancers-14-04881-f002:**
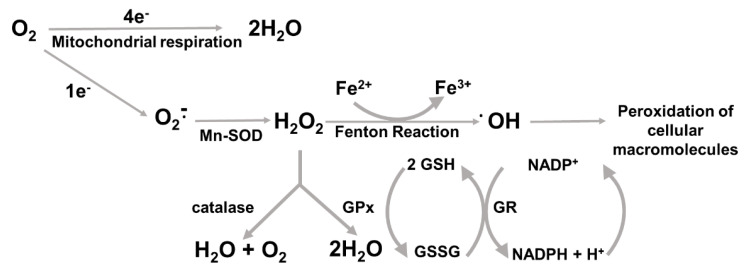
Major sources and reactions underlying the generation of ROS and the early cytoprotective systems detoxifying ROS. Mn-SOD, manganese-dependent superoxide disutase; GSH, reduced glutathione; GSSG, glutathione disulfide; GPx, glutathione peroxidase; GR, glutathione reductase; NADP+/NADPH, oxidized/reduced nicotinamide adenine dinucleotide phosphate.

**Figure 3 cancers-14-04881-f003:**
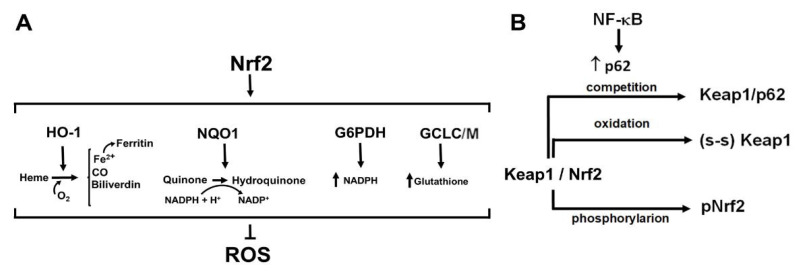
Activation and function of Nrf2. (**A**) Nrf2 activation induces the expression of a wide variety of antioxidant factors. (**B**) Keap1-mediated inhibition of Nrf2 can be relieved by competition of p62 with Keap1, by oxidation of cysteines harbored within the Keap1 sequence, or phosphorylation of Nrf2, especially by PKC, enabling Nrf2 to enter the nucleus and bind to AREs. (s-s) Keap1, oxidized Keap1; HO-1, heme oxygenase 1; NQO1, NAD(P)H dehydrogenase [quinone] 1; G6PDH, glucose-6-phosphate dehydrogenase; GCLC/M, glutamate-cysteine ligase (GCL) catalytic (C) and modulatory (M) subunits.

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
