# Peer review of "Protein Phosphorylation and Redox Status: An as Yet Elusive Dyad in Chronic Lymphocytic Leukemia"

_cancers, 2022, doi:10.3390/cancers14194881_

Round 1
Reviewer 1 Report (New Reviewer)
They could have talked about linking TKIs and ROS levels in CLL. BTK inhibitors are the most successful in the clinic, does it modulate ROS levels in CLL or is there a link?
Author Response
Reviewer 1
They could have talked about linking TKIs and ROS levels in CLL. BTK inhibitors are the most successful in the clinic, does it modulate ROS levels in CLL or is there a link?
At the end of Chapter 7, we mentioned the generation of ROS by using ibrutinib and venetoclax, the mechanism of action of which is still under investigation. At present, it is known that ibrutinib can induce activation of NOX in neutrophils, but NOX is not abundantly expressed in CLL cells, having a marginal role in ROS production.
Reviewer 2 Report (New Reviewer)
This is an interesting review on the connection between redox activity and the phosphorylation of pivotal proteins in CLL ( an likely many other cell types). I agree with reviewer #1 that the paper is interesting but to a specialized audience. I only have minor comments
1.- The points 1-3 are easy to follow , anad thet the paper deals with Nrf2, NF-lB and FOXO transcription factors and their phosphorylation modifications due to redox activities. The three paragraphs are interetisnt but the connection to CLL must be emphasized. For example, the expression levels of these factors in CLL must be clearly described in the initial sentences of the corresponding part (4, 5, 6).
3.- At least some of the known transcriptional target nesf Nfr2 should be cited in part 4.
4.- The part between lines 169-191 are difficult to follow . A figure similar to Figure 1 will be very helpful
5.-PKC is referrd to as a single proteins bu in fact there are several PKCs
6.- Line 168: p21 is a well-known p53 target gene, but I do not see why to mention p53 here
Author Response
Reviewer 2
This is an interesting review on the connection between redox activity and the phosphorylation of pivotal proteins in CLL ( an likely many other cell types). I agree with reviewer #1 that the paper is interesting but to a specialized audience. I only have minor comments
1.- The points 1-3 are easy to follow , anad thet the paper deals with Nrf2, NF-kB and FOXO transcription factors and their phosphorylation modifications due to redox activities. The three paragraphs are interetisnt but the connection to CLL must be emphasized. For example, the expression levels of these factors in CLL must be clearly described in the initial sentences of the corresponding part (4, 5, 6)
We added a few lines regarding the expression level of the individual transcription factors.
3.- At least some of the known transcriptional target nesf Nfr2 should be cited in part 4
We changed the introduction of the chapter on Nrf2, emphasizing the role of Nrf2 as such and referring the readership to Figure 3A, and reporting the main transcriptional targets of Nrf2 in CLL
4.- The part between lines 169-191 are difficult to follow . A figure similar to Figure 1 will be very helpful
We totally re-wrote the chapter on Nrf2 in hopes that it is now clearer. As the role of these kinases, albeit known in other context, is still speculative in CLL cells, we surmise that a Figue might be misleading for the readership.
5.-PKC is referrd to as a single proteins bu in fact there are several PKCs
In the chapter illustrating Nrf2, We added the subfamilies of PKC.
6.- Line 168: p21 is a well-known p53 target gene, but I do not see why to mention p53 here
We just mentioned p53 in relation to its transcriptional target with regard to physiological conditions, as I did for p62, which is instead closely related to NF-kB and CLL. We also rephrased the sentence.
This manuscript is a resubmission of an earlier submission. The following is a list of the peer review reports and author responses from that submission.
Round 1
Reviewer 1 Report
This review manuscript is highlighting a general overview of oxidative stress and phosphorylation-dependent signals in CLL. This paper is very interesting and on a timely subject. The idea that oxidative stress has a pivotal role in cancer initiation, progression, and resistance to therapy is now widely accepted. However, in CLL it is still a matter of debate. Overall, this is a well-written and detailed manuscript with literature research.
- I suggest writing Figure 1 and 3 legends so that the reader can fully understand the content of your figure without having to refer to the main text.
- Make sure abbreviations are consistent with the text of the paper, eg. (s-s)Keap1 (Figure 3A).
- Point 4. Nrf2, the master regulator of ... - line 250 - Figure 3 should be corrected by adding B (should be Figure 3B).
- Moreover, only minor language corrections should be necessary.
Author Response
Referee # 1
I suggest writing Figure 1 and 3 legends so that the reader can fully understand the content of your figure without having to refer to the main text.
Reply
The legends have been properly modified.
Make sure abbreviations are consistent with the text of the paper, eg. (s-s)Keap1 (Figure 3A).
Reply
Amended.
Point 4. Nrf2, the master regulator of … – line 250 – Figure 3 should be corrected by adding B (should be Figure 3B).
Reply
Amended
Moreover, only minor language corrections should be necessary.
Reply
Text thoroughly revised
Reviewer 2 Report
Pagano and colleagues present a review of the role of reactive oxygen species (ROS) and protein phosphorylation signals in chronic lymphocytic leukemia (CLL). They introduce the reader to B-cell receptor signaling, the antioxidant response, and regulators of oxidative stress. The authors suggest that ROS may be a potential drug target in CLL, and that such treatments may synergize with currently administered targeted therapies. The topic of the review is expected to be of interest to the specialized audience.
The ‘simple summary’ and ‘abstract’ of the manuscript are very well presented and trigger interest in the review article. However, there are some concerns regarding the main text, especially when it comes to language and style:
- Many of the sentences in the manuscript are very long. This makes it difficult at times to capture the whole message of the sentence. Examples can be found on pages 2-3 of the manuscript:
‘The most investigated receptor of B cells, which is involved not only in the immune response but also in the activation of parallel pathways regulating survival, maturation, migration of the B cell itself, to name a few, is the B cell receptor, composed by an immunoglobulin whose role is to recognize antigens, and the co-receptors CD79a and CD79b, the signaling components of the BCR [17].’
‘Src Family Kinase (SFK) predominantly expressed in B lymphocytes, which phosphorylates specific tyrosines within the immunoreceptor tyrosoine-based activation motifs (ITAMs) of a variety of co-receptors, namely CD79a and CD79b themselves and CD19, thereby providing the docking sites for the SH2 domain-based recruitment, and subsequent activation, of the effector kinases Syk and Phosphoinositide-3-kinase (PI3K) [33].’
Sentences of this extensive format need to be re-formulated to increase readability.
- Some sentences contain so much information it becomes difficult to understand what the authors want to say. One example can be found on page 8:
‘Whereas there is evidence that IKKb can be inhibited by oxidation at cysteine 179, not only as an effect of severe oxidative stress but also as a negative feed-back response under physiological conditions [110], the regulatory subunit NEMO can undergo oxidation of Cys54 and Cys347, with subsequent formation of a disulfide bond and dimerization that ultimately triggers IKK activity [111] also bearing in mind that the inhibitory effect of ROS on phosphatases regulating IKK cannot be ruled out [112].’
- The manuscript contains grammatical errors and spelling mistakes that should be corrected. Importantly, ‘microenvironment’ needs to be corrected throughout.
- Abbreviated protein names are not consistently introduced. For example, BTK (Bruton’s Tyrosine Kinase) is introduced on line 47, while numerous other proteins, such as Lyn, Syk, and Akt (line 67), are not spelled out. BTK is introduced again on line 109, and is interchangeably referred to as Btk and BTK (I cannot see that this refers to gene name and product, respectively). Please make this consistent.
- Figure 2 shows gridlines and underlines that probably come from the software the figure was made in, and presumably should not appear in the final version of the figure.
- In the ‘simple summary’, the authors mention ROS as a therapeutic target. It would be of interest if the authors could elaborate more on this in the main text, and better indicate at what stage the drug development is. What molecules are targeted, are there any ongoing clinical trials etc.
In conclusion, I think the topic of the review is of potential interest. However, in its present form, the readability of the manuscript is compromised by very long sentences and unclear formulations.
Author Response
(Comments to the Author):
Many of the sentences in the manuscript are very long. This makes it difficult at times to capture the whole message of the sentence. Examples can be found on pages 2-3 of the manuscript:
“The most investigated receptor of B cells, which is involved not only in the immune response but also in the activation of parallel pathways regulating survival, maturation, migration of the B cell itself, to name a few, is the B cell receptor, composed by an immunoglobulin whose role is to recognize antigens, and the co-receptors CD79a and CD79b, the signaling components of the BCR [17].”
“Src Family Kinase (SFK) predominantly expressed in B lymphocytes, which phosphorylates specific tyrosines within the immunoreceptor tyrosoine-based activation motifs (ITAMs) of a variety of co-receptors, namely CD79a and CD79b themselves and CD19, thereby providing the docking sites for the SH2 domain-based recruitment, and subsequent activation, of the effector kinases Syk and Phosphoinositide-3-kinase (PI3K) [33].”
Sentences of this extensive format need to be re-formulated to increase readability.
Reply
The text was thoroughly revised, the sentences were shortened, hopefully with better readability
Some sentences contain so much information it becomes difficult to understand what the authors want to say. One example can be found on page 8:
“Whereas there is evidence that IKKb can be inhibited by oxidation at cysteine 179, not only as an effect of severe oxidative stress but also as a negative feed-back response under physiological conditions [110], the regulatory subunit NEMO can undergo oxidation of Cys54 and Cys347, with subsequent formation of a disulfide bond and dimerization that ultimately triggers IKK activity [111] also bearing in mind that the inhibitory effect of ROS on phosphatases regulating IKK cannot be ruled out [112].”
Reply
The text was thoroughly revised, the sentences were shortened, hopefully with better readability
The manuscript contains grammatical errors and spelling mistakes that should be corrected. Importantly, ‘microenvironment’ needs to be corrected throughout.
Reply
Revised and amended
Abbreviated protein names are not consistently introduced. For example, BTK (Bruton’s Tyrosine Kinase) is introduced on line 47, while numerous other proteins, such as Lyn, Syk, and Akt (line 67), are not spelled out. BTK is introduced again on line 109, and is interchangeably referred to as Btk and BTK (I cannot see that this refers to gene name and product, respectively). Please make this consistent.
Reply
Amended
Figure 2 shows gridlines and underlines that probably come from the software the figure was made in, and presumably should not appear in the final version of the figure.
Reply
Amended
In the ‘simple summary’, the authors mention ROS as a therapeutic target. It would be of interest if the authors could elaborate more on this in the main text, and better indicate at what stage the drug development is. What molecules are targeted, are there any ongoing clinical trials etc.
Reply
A short paragraph was added before Conclusions
Round 2
Reviewer 2 Report
My comments from the first review still apply, and I cannot recommend publication of the manuscript in its present form.